# Ultrasonic Induced Refinement of Induction Heated Oxide Coating on Titanium

**Han Gao** [1,2], **Meijie Yu** [1,2], **Xin Chen** [1,2], **Guiyong Xiao** [1,2], **Chuanzhong Chen** [1,2], **Bing Liu** [1,2,3,*] and **Yupeng Lu** [1,2,*]

1 Key Laboratory for Liquid-Solid Structural Evolution and Processing of Materials, Ministry of Education, Shandong University, Jinan 250061, China; 201713597@mail.sdu.edu.cn (H.G.); yumeijie@sdu.edu.cn (M.Y.); 201612521@mail.sdu.edu.cn (X.C.); xiaoguiyong@sdu.edu.cn (G.X.); czchen@sdu.edu.cn (C.C.)
2 School of Materials Science and Engineering, Shandong University, Jinan 250061, China
3 Suzhou Institute of Shandong University, Suzhou, 215123, China
* Correspondence: liubingicy@sdu.edu.cn (B.L.); biosdu@sdu.edu.cn (Y.L.); Tel.: +86-187-5314-1628 (B.L.); +86-138-5318-6026 (Y.L.)

**Abstract:** Induction heating treatment (IHT) has recently been used to improve the bioactivity and biocompatibility of titanium and its alloys, greatly related to the formation of the nanoscale oxide coating. In this work, the effect of ultrasonic on the IHT oxidation behavior of pure titanium has been investigated. Ultrasonic-assisted IHT of pure titanium was carried out for 13, 20 and 25 s. Submicro-/nano-scale morphological coatings with rutile and anatase $TiO_2$ were prepared on the surface of titanium substrates subjected to ultrasonic-assisted IHT. In particular, the $TiO_2$ crystals were significantly refined by ultrasonic impact. An improvement in hydrophilicity and hardness of the oxide film was achieved by ultrasonic-assisted IHT. The refinement of $TiO_2$ crystals is suggested to be caused by ultrasonic induced changes of energy, defect density and their correlation with diffusion of oxygen. The present study provides a potential method to refine the nanoscale oxide films on titanium substrates, which is promising for improving the wear resistance and bioactivity of titanium and its alloys.

**Keywords:** titanium; induction heating treatment; ultrasonic; grain refinement

## 1. Introduction

Titanium and its alloys are widely used in biomedical implant field due to their attractive properties including high strength, good biocompatibility, excellent corrosion resistance and moderate elastic modulus [1]. However, the bioactivity of titanium surface needs to be improved to achieve early osteointegration. Cell-to-cell, cell-to-protein, and cell-to-biological tissue interactions, such as surface sensing and recognition as well as signal transfer, occur at the molecular level in nanoscale [2–5]. The nanoscale feature of material surface has significant influence on cell adhesion, migration, proliferation, differentiation and apoptosis both in vitro and in vivo [6,7]. In addition, the nanoscale feature on implant surface can promote the adsorption of proteins and stimulate the osteogenic cell migration, thus leading to rapid osseointegration [8,9]. As a result, adding nanostructure on implant surface is expected to enhance osteoconductivity and early-stage osseointegration [10]. On the other hand, the wear resistance of titanium-based biomaterials requires special attention [11]. For example, when the implant made of Ti6Al4V alloy is in contact with other metals, polyethylene or bone, the abrasion induced by relative motion at the interface can weaken the passivated titanium oxide, which results in debris and metal ions releasing.

The controlled oxidation is an attractive method for the preparation of micro-/nano-textured topography. Moreover, oxidation has been proved to be an efficient approach to improve the wear resistance of titanium and its alloys [12–14]. Induction heating treatment (IHT) has attracted more attention to produce oxide coating on titanium, due to its fast

heating, reduced heat loss, cleanness and environmental safety [15]. During IHT process, due to the skin effect of current, the temperature of the surface increases rapidly, while the temperature of the inner part rises at a much lower rate [16]. As a result, a layer of titanium oxide coating forms rapidly on the surface while the intrinsic mechanical properties of the substrate remain constant. Markovsky and Semiatin [17] found that the tension and fatigue properties obtained by IHT and aging of Ti6Al4V with an initial microstructure of fine-grain equiaxed α phase are comparable to the optimal one obtained by rapid bulk heat treatment of the same alloy. IHT has recently been used to improve the bioactivity and biocompatibility of titanium and its alloys. Fomin et al. [18,19] preliminarily prepared titanium dioxide coatings with nanocrystalline structure by IHT and demonstrated the improved biocompatibility of the involved medical titanium. Li et al. [16,20] also reported the use of IHT to produce titanium dioxide coatings with a micro- and nano- crystalline structure and the improved hydroxyapatite (HA) deposition in simulated body fluid (SBF). We would like to note that, special attention is paid to the enhancement of osteointegration process by improving the morphology and geometrical characteristics, which depend primarily on the duration and temperature in the above-mentioned IHT process.

Ultrasonic energy is mechanical energy and the propagation of the ultrasonic wave in the solids can greatly affect the microstructure of the material [21,22]. Ultrasonic energy is considered to be absorbed preferentially by lattice defects like dislocations and grain boundaries, resulting in reduced activation energy for dislocation to overcome obstacles and enhanced dislocation mobility [23]. Acoustic softening and acoustic residual hardening effects were modeled based on defect evolution [24,25]. Ao et al. [26] reported that the gradient nanostructure of the β phase in Ti6Al4V alloy subjected to ultrasonic surface rolling process was formed primarily via dislocation activities. The numerous grain boundaries and dislocations of titanium surface can accelerate the adsorption of oxygen including the physical adsorption of oxygen molecules followed by dissociation, chemisorption and reaction [27]. Moreover, the increased defects with loose atomic arrangement and high energy facilitate the transformation of ions and electrons during oxidation and then the preferential nucleation and crystallization of $TiO_2$ crystallites [28]. Zhang et al. [29] demonstrated that the acceleration of oxidation at initial stage was mainly due to dislocation accumulation. The ultrasonic-assisted IHT process has been used by Osada et al. [30] for bonding aluminum structure, via plastic flow in the aluminum structure softened by ultrasonic energy and disruption of the oxide film.

To the best of our knowledge, no research about ultrasonic-assisted IHT to control oxide structure has been reported so far. The objective of this work is to study the effects of ultrasonic, as a kind of high frequency low amplitude mechanical wave, on the IHT oxidation behavior of titanium, including phase and microstructural evolution during IHT.

## 2. Materials and Methods

### 2.1. Sample Preparation

Samples with dimensions of $10 \times 20 \times 2$ mm$^3$ were cut from a commercially pure titanium (cp-Ti, grade 2) plate using wire electrical discharge machining. Then they were successively abraded with silicon carbide papers from grade 400 to 1000, and then cleaned in an ultrasonic cleaner with acetone, ethanol and deionized water, respectively.

### 2.2. IHT Process of Ti Specimens with Ultrasonic Impact

Ultrasonic-assisted IHT was carried out on an apparatus as shown schematically in Figure 1. The specimen was clamped onto the upper surface of a transducer with a power of 50 W. Ultrasonic vibration was provided by an ultrasonic generator working at 28 kHz. The transducer connected to the generator converted the signal to unidirectional mechanical vibration onto the specimen. The transducer/specimen assembly was then placed underneath a plane inductor with a frequency of 50 kHz and a power level of 50 kW. The other group prepared without ultrasonic assistance was used as control. The samples in each group were heated separately for 13, 20 and 25 s and the temperature

reached 750 °C, 800 °C and 900 °C, respectively. Then the samples were cooled in air. The samples were recorded as U13s, U20s and U25s for ultrasonic-assisted group and C13s, C20s and C25s for the control. The sample without subjecting to IHT process was recorded as Machined.

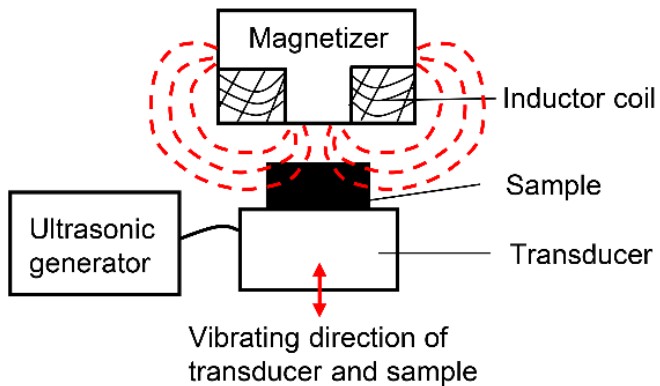

**Figure 1.** Schematic diagram of the experimental set.

*2.3. Phase and Microstructural Characterization*

X-ray diffraction (XRD, DMAX-2500PC, Rigaku, Tokyo, Japan) was used to identify the phase compositions of the samples, which was performed using Cu Kα radiation with a wavelength of 0.154 nm. The goniometer running at 50 kV and 100 mA was used to scan a 2θ range from 10° to 90° with a scan rate of 10°/min. The surface morphologies of the samples were observed by a field emission scanning electron microscope (FE-SEM, JSM-7800F, JEOL, Tokyo, Japan). The element distribution was detected by energy dispersive X-ray spectroscopy (EDS, Oxford XMax-80, Oxford, UK) integrated with SEM. The 3D topography and roughness were measured using laser scanning confocal microscopy (LSCM, LSM-800, Zeiss, Oberkochen, Germany) within a horizontal scanned area of $600 \times 600 \ \mu m^2$. The laser wavelength was 405 nm and the root mean squared roughness (Sq) and arithmetic average roughness (Sa) were calculated from the scanned area using the Zen Blue software. The cross-sectional microstructure and element distribution were examined using electron probe microanalysis (EPMA, JXA-8530F Plus, JEOL, Tokyo, Japan). The acceleration voltage was 10 kV and the electron beam current was 30 nA.

*2.4. Contact Angle Measurement*

Contact angle measurements were done through the sessile drop method using a contact angle instrument (DSA100S, Kruss, Hamburg, Germany). A microliter syringe tip was used to put a 2 μL droplet of distilled water on the surface of each sample which was pre-adjusted to a certain height to ensure that the droplet made just enough contact with the surface of the sample. The drop image was recorded by a video camera and processed by an image analysis software.

*2.5. Micro-Hardness Test*

The surface micro-hardness (HV) test was performed using a Micro Vickers Hardness Tester (DHV-1000, Caikon, Shanghai, China) with a diamond indenter at ambient condition. A static load of 100 g for 10 s was applied to each specimen surface. For each specimen, the average of five indentations was used for the statistical analysis to ensure acquisition of reasonably representative value.

**3. Results and Discussion**

*3.1. Phase Composition*

The XRD patterns of cp-Ti specimens treated by IHT with and without ultrasonic assistance are shown in Figure 2. Some diffraction peaks in the C25s group are covered, so an enlarged diagram is inserted in Figure 2. After IHT for 13–25 s, the diffraction peaks

from titanium substrate are obviously observed from the XRD spectra. After oxidation for 13 s, rutile phase can be found in both the control and the ultrasonic-assisted sample. After oxidation for 20 s and 25 s, the XRD patterns exhibit the presence of rutile phase and a small amount of anatase phase. Moreover, with the increase in IHT period from 13 s to 25 s, the rutile mainly tends to grow along <110> plane at 2θ degree of 27.446°. After oxidation for 25 s, the diffraction intensity from the substrate is obviously lower in the ultrasonic-assisted sample than that of the control, implying the ultrasonic-enhanced oxidation. During thermal oxidation, rutile $TiO_2$ is the prevalent oxide type at and above 600 °C [31] while the formation of anatase phase usually occurs at lower treatment temperatures and time due to its metastability in nature [32]. The transition from anatase to rutile has not completed during rapid oxidation, which may be the reason for the presence of anatase after IHT for 20 s and 25 s. The bioactivity of the $TiO_2$ film is in close correlation with the phase structure. $TiO_2$ with a rutile or anatase structure enables to deposit apatite on its surface in SBF, and anatase phase is more effective for apatite formation [33] and yield the better biological effects for cell adhesion, spreading, proliferation and differentiation [34] compared to rutile phase. On the other hand, rutile phase shows a better corrosion resistance [35].

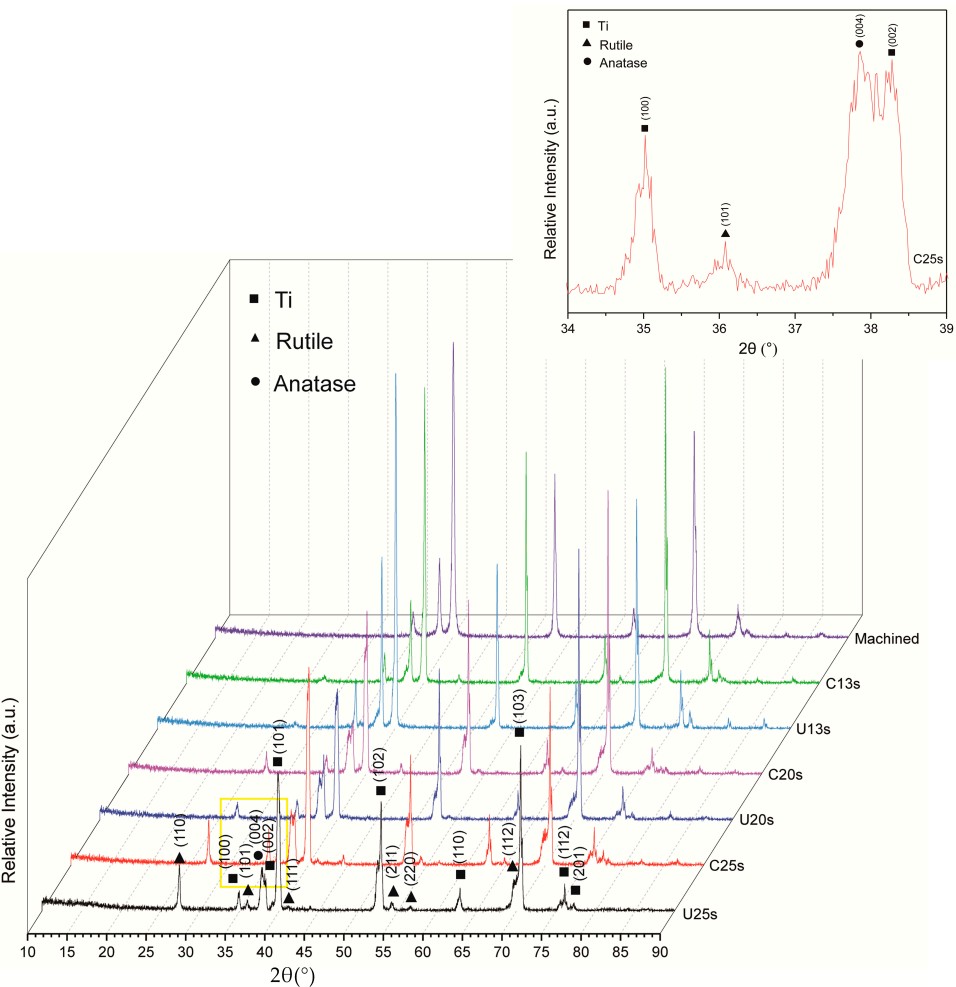

**Figure 2.** XRD patterns of titanium surface before and after IHT with and without ultrasonic assistance for 13–25 s. The insert diagram is an enlarged diagram for some diffraction peaks (in the yellow wireframe) in the C25s.

### 3.2. Surface Morphology and Element Distribution

The SEM images of the surface oxide layer and the corresponding atomic ratios of oxygen to titanium are shown in Figure 3. Numerous grain-like $TiO_2$ nano- and submicro-scale crystals are formed on the surfaces of cp-Ti samples subjected to IHT for 13–25 s.

Obviously, the increase of IHT time results in the increase in the sizes of $TiO_2$ particles, which is due to the higher thermal energy and enough time for grain growth. At the same time, the sizes of $TiO_2$ particles are obviously decreased when subjected to IHT process with ultrasonic assistance. The average sizes of $TiO_2$ particles decrease from 60 nm, 70 nm and 150 nm in the control to 10 nm, 40 nm and 100 nm in the ultrasonic-assisted samples after IHT for 13 s, 20 s and 25 s, respectively. Ion doping has been reported to reduce the $TiO_2$ particle size due to lattice strain induced by a mismatch of the ionic radii of $Ti^{4+}$ and doped ions [36]. Mitchell et al. [37] reported the significant grain refinement of atomic layer deposition $TiO_2$ films due to the increased nucleation sites for crystallization and growth stress. The nanostructure composed by anatase and rutile $TiO_2$ have been demonstrated to exhibit the improved cell adhesive and protein adsorption ability [38]. Nanofeature-enhanced osteoconductivity, which results in both the acceleration and elevation of bone-implant integration, has been clearly clarified [10]. It can be concluded that the nanoscale $TiO_2$ film obtained by ultrasonic-assisted IHT will perhaps accelerate the osteointegration of the surface.

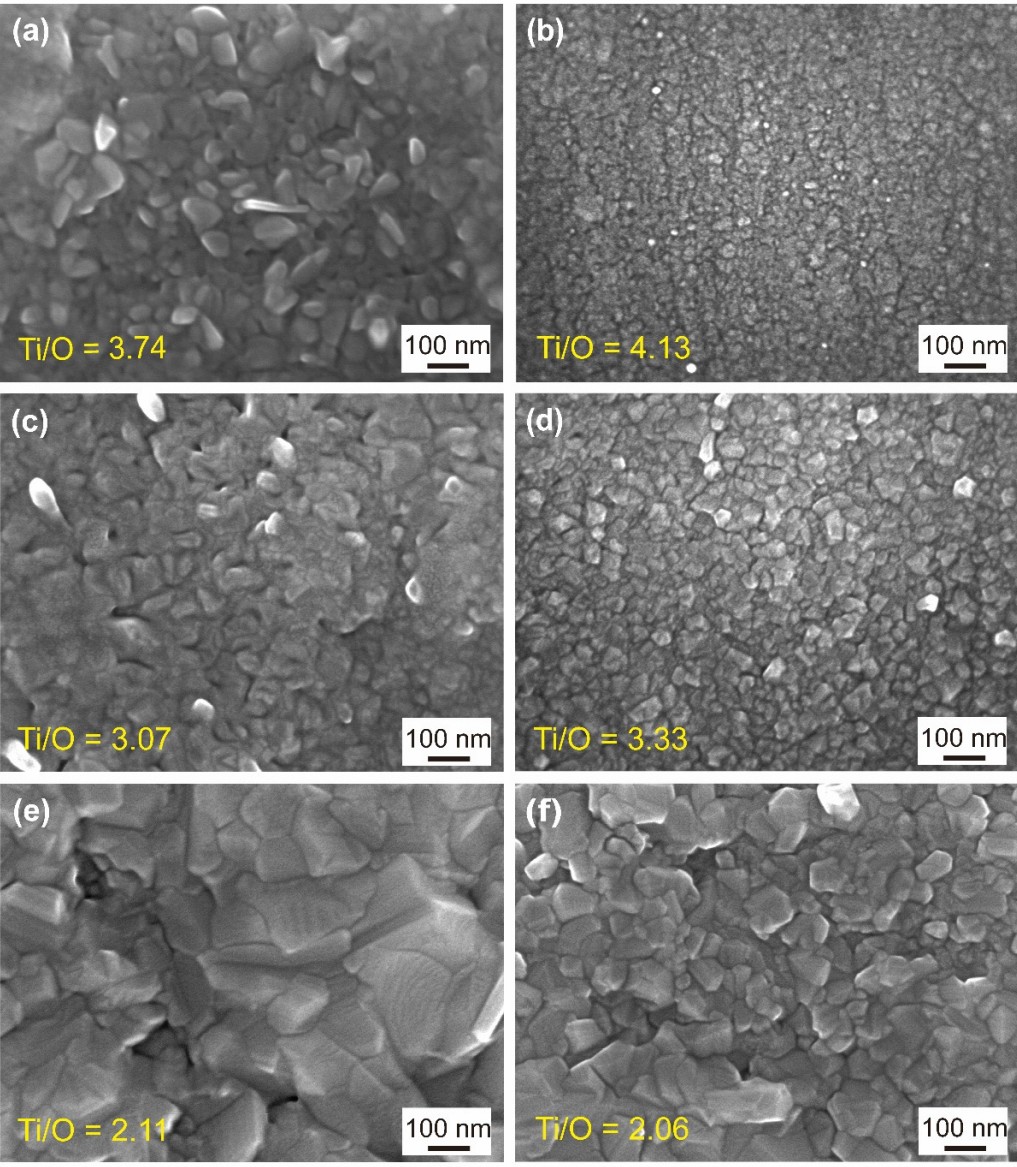

**Figure 3.** SEM images of titanium surface after IHT for 13 s (**a**), 20 s (**c**) and 25 s (**e**), ultrasonic-assisted IHT for 13 s (**b**), 20 s (**d**) and 25 s (**f**).

Closer examination of the element content of oxygen and titanium on the oxide layer surfaces was carried out based on EDS measurements. The oxygen contents and their increment increase as the IHT period is upregulated. The oxygen content of the control is higher than that of the ultrasonic-assisted sample after IHT for 13 and 20 s, while the atomic ratios of oxygen to titanium show no apparent difference between the ultrasonic-assisted specimen and the control after IHT for 25 s.

In order to determine the characteristic topographical features and surface roughness of the oxide layer, LSCM was performed within a scanned area of $600 \times 600 \ \mu m^2$ and the corresponding results are presented in Figures 4 and 5. The grooves formed by polish can be clearly observed in Figure 4a. After IHT for 13–25 s, the 3D images show obvious abrasive scratches yet, since the $TiO_2$ film is fairly thin. After IHT for 13 s, due to the formation of $TiO_2$ particles, the Sq and Sa increase to 171 and 115 nm in the control, 144 and 110 nm in the ultrasonic-assisted specimen, respectively, compared with that of the machined specimen (82.7 and 58.7 nm). With the prolongation of IHT time, more $TiO_2$ particles continuously grow up and fill the grooves, leading to the decrease of Sq and Sa to 121 and 95.1 nm in the control, 102 and 75 nm in the ultrasonic-assisted specimen after IHT for 20 s, and 121 and 88.1 nm in the control, 80.9 and 60.8 nm in the ultrasonic-assisted specimen after IHT for 25 s, respectively. Moreover, the roughness of the ultrasonic-assisted samples is lower than that of the control. Omidbakhsh et al. [39] reported that the Sa of Ti4Al2V increased with time and temperature of oxidation and increased from 20 to 130 nm after 1 h oxidation at 700 °C. The surface roughness of implants affects the rate of osseointegration and biomechanical fixation and nanometer roughness plays an important role in the adsorption of proteins, adhesion of osteoblastic cells and thus the rate of osseointegration [40]. According to Huang et al. [41], the ground titanium specimen with an Sa value of 150 nm has the best cell adhesion and spreading appearance compared with either the smoother (Sa: 50 and 70 nm) or rougher (Sa: 330 and 1200 nm) specimens. Similar results have been reported by Keller [42], a tight adhesion of cells on a sandpaper-ground Ti specimen (Sa: 100 nm). Hence, it may be concluded that a lower IHT time with ultrasonic assistance is required for the suitable surface roughness to increase cell adhesion.

### 3.3. Cross-Sectional Morphology and Element Distribution of the Oxide Layer

The cross-sectional morphologies of the specimens subjected to IHT for 25 s and the corresponding element distribution of oxygen and titanium from the top surface to the bulk of the specimens are shown in Figure 6. After IHT for 25 s, the oxide layers formed on both the control and ultrasonic-assisted specimen are homogeneous and exhibit an excellent adherence to the substrate (Figure 6a,b). The oxygen contents decrease with the increase of distance away from the top surface of both the control and ultrasonic-assisted specimen. The thicknesses of the oxide layers formed on the specimens treated by IHT with and without ultrasonic assistance are about 1.2 (Figure 6d) and 1.6 μm (Figure 6c), respectively. Liu et al. [43] reported the enhanced adsorption and reaction capability of Ti6Al4V with oxygen at 500 and 600 °C, because of the numerous dislocations and grain boundaries introduced by ultrasonic nanocrystal surface modification. In this study, the rapid formation of oxide coating may prevent the further oxidation during ultrasonic-assisted IHT process, therefore the oxide layer of the ultrasonic-assisted specimen is thinner than that of the control.

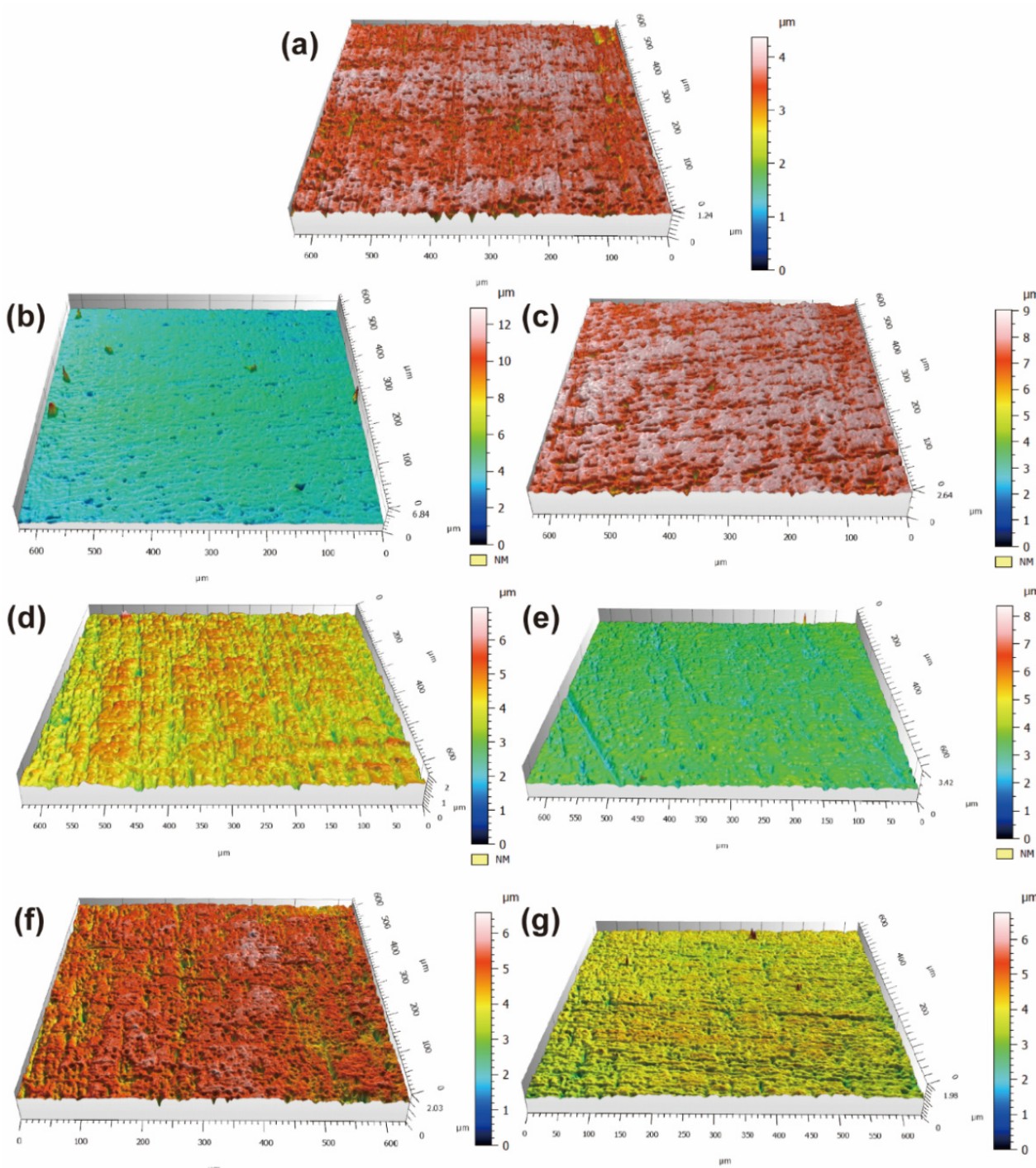

**Figure 4.** 3D plot of the topography on titanium (**a**) after IHT for 13 s (**b**), 20 s (**d**) and 25 s (**f**), ultrasonic-assisted IHT for 13 s (**c**), 20 s (**e**) and 25 s (**g**).

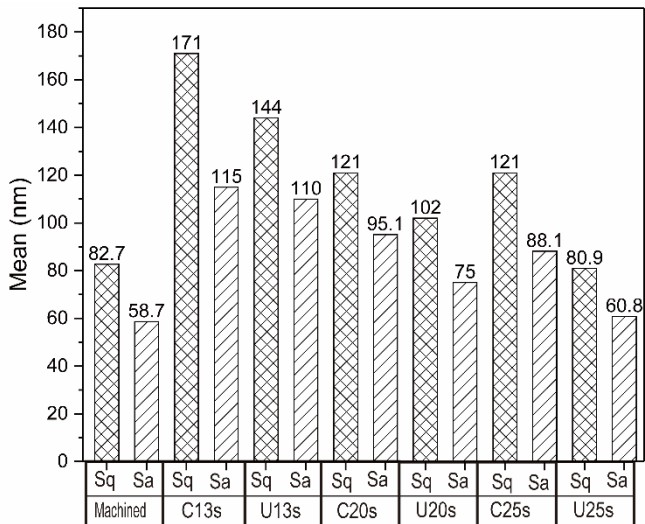

**Figure 5.** Root mean squared roughness (Sq) and arithmetic average roughness (Sa) on titanium (Machined) after IHT for 13 s (C13s), 20 s (C20s) and 25 s (C25s), ultrasonic-assisted IHT for 13 s (U13s), 20 s (U20s) and 25 s (U25s).

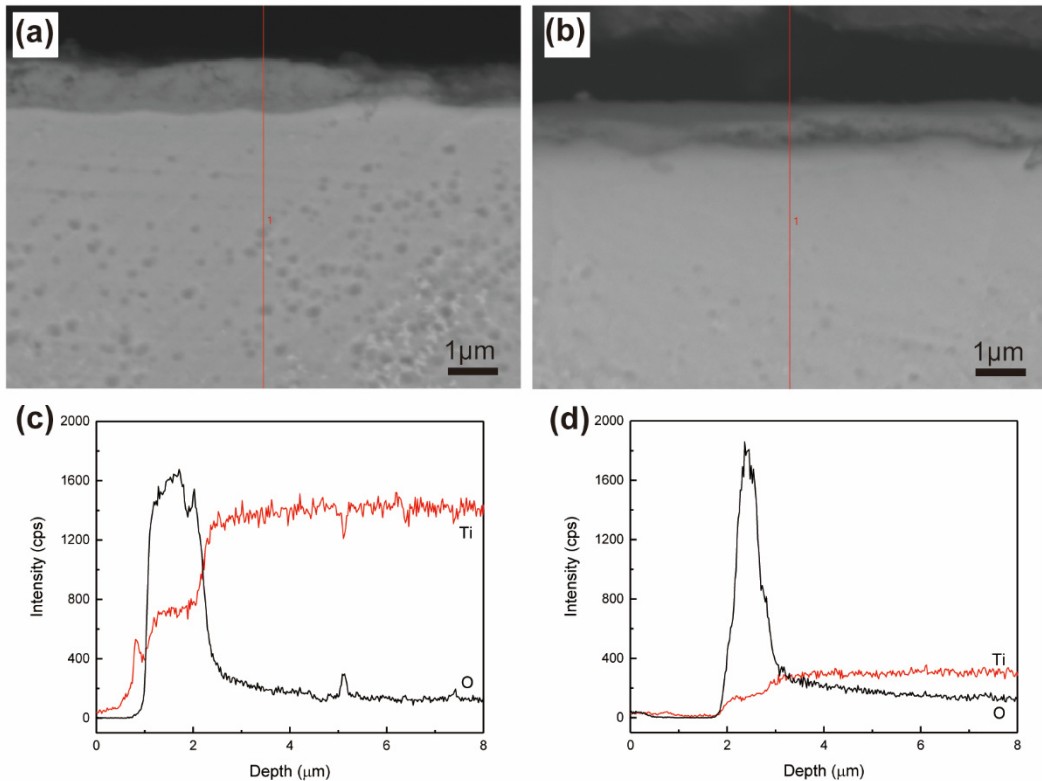

**Figure 6.** Cross-sectional EPMA images of titanium after IHT for 25 s (**a**) and ultrasonic-assisted IHT for 25 s (**b**). EPMA line scanning patterns of oxygen and titanium across the oxidation layer in titanium after IHT for 25 s (**c**) and ultrasonic-assisted IHT for 25 s (**d**).

### 3.4. Surface Wettability

Figure 7 shows the images of the water droplet on the modified surfaces and the corresponding contact angles. After IHT for 13 s, the control exhibits lower contact angle than the machined sample (82.2°), while the ultrasonic-assisted sample exhibits higher contact angle. After IHT for 20 s, both the control and ultrasonic-assisted sample exhibit the

largest contact angle. With the continuous prolongation of IHT time, the contact angles of the control and ultrasonic-assisted sample reduce to 68.8° and 62°, respectively, indicating that IHT as well as ultrasonic plays an important role in increasing the wettability within a certain period. It is found that the thermal oxidation time influences the wettability of titanium implants [44] and Li et al. [16] reported that the contact angle decreased successively to 63.9° ± 1.7° upon increasing IHT time to 35 s. There is optimal cell adhesion to moderately hydrophilic substrates, due to the adsorption of cell adhesion-mediating molecules (e.g., vitronectin, fibronectin) in an advantageous geometrical conformation, which makes specific sites on these molecules (e.g., specific amino acid sequences) accessible to cell adhesion receptors (e.g., integrins) [45]. Surfaces with higher wettability are likely to adhere more cells [46] and Heloisa et al. reported that the anatase and rutile mixture film with contact angle of 34° exhibited great protein adsorption and cell adhesion and spreading [47]. Therefore, the increase in surface wettability following ultrasonic-assisted IHT will perhaps increase the cell adhesion.

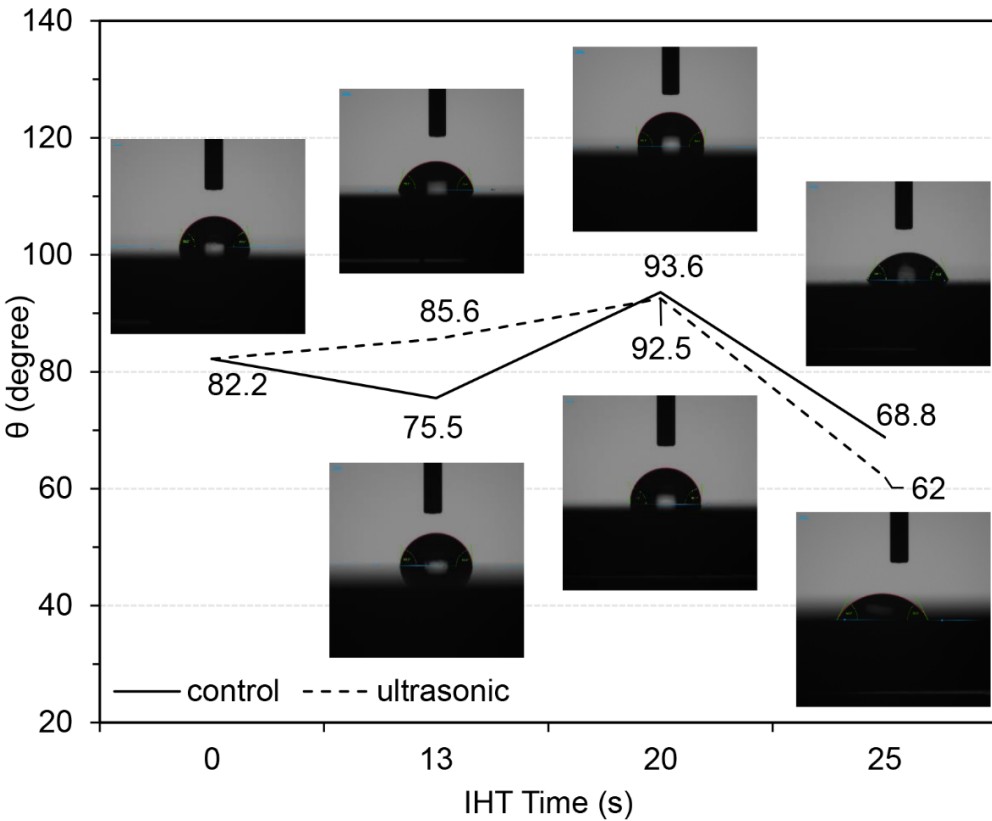

**Figure 7.** The water droplet on the titanium surface and the contact angles before and after IHT with and without ultrasonic assistance for 13–25 s.

*3.5. Micro-Hardness*

The micro-hardness of cp-Ti specimens before and after IHT is shown in Figure 8. It can be found that the surface hardness is improved by IHT compared to the machined cp-Ti (212.0 ± 6.2 HV). In addition, the hardness is further improved with the continuous prolongation of IHT time. After IHT for 25 s, the hardness of the samples treated by IHT with and without ultrasonic assistance increase to 393.6 ± 13.4 HV and 376.8 ± 6.2 HV, respectively. According to Li et al. [20], the hardness of cp-Ti increased by about 25 HV after cold-drawing with 10–20% deformations and following IHT for 20 s. Generally, thermal oxidation of titanium enables an improvement of hardness due to the formation of a hard oxide layer and the presence of an oxygen diffusion zone beneath it [48,49]. On the other hand, there is no statistically significant difference between the control and ultrasonic-assisted samples, indicating that the influence of ultrasonic on surface hardness

can be negligible. The improvement of surface hardness could effectively improve the wear resistance and fatigue strength of titanium implants, thus reduce the peeling of wear debris after implanted in vivo, and reduce the occurrence of inflammation and the failure rate of implantation surgery [50].

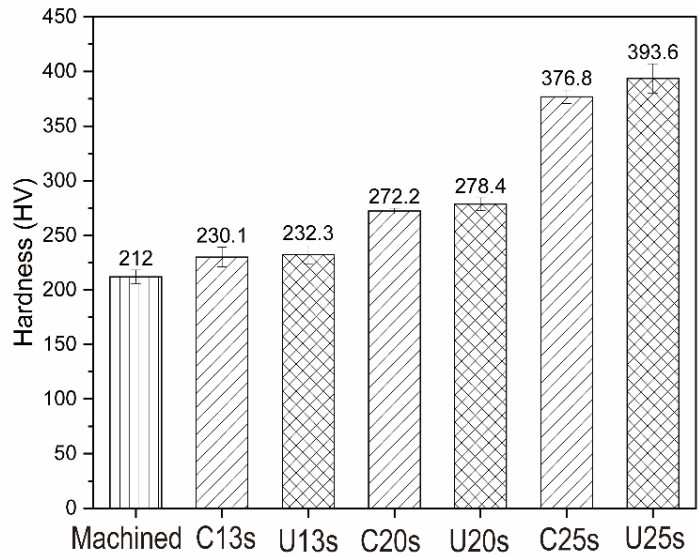

**Figure 8.** Micro-hardness of titanium surface (Machined) after IHT for 13 s (C13s), 20 s (C20s) and 25 s (C25s), ultrasonic-assisted IHT for 13 s (U13s), 20 s (U20s) and 25 s (U25s). The data is shown as mean ± SD, n = 5.

### 3.6. Mechanism of Ultrasonic Induced Refinement of Induction Heated Oxide Coating on Titanium

As described above, ultrasonic adopted in IHT of cp-Ti leads to the obviously refined $TiO_2$ particles on its surface and resultant improved wettability and hardness. The refinement of $TiO_2$ particles is suggested to be caused by ultrasonic induced changes of energy, defect density and their correlation with diffusion of oxygen.

The oxidation of metals comprises three main stages [27,51], i.e., dissociative chemisorption of oxygen on the surface, incorporation of oxygen atoms into the lattice of a metal and development of the surface oxide layer. The density of dislocation and other structural defects increases under ultrasonic impact [26], which affects all the stages, so the mechanism of ultrasonic induced refinement of induction heated oxide coating on titanium can be conceived, as shown in Figure 9. Firstly, the various high-density structural defects generated by ultrasonic impact on the metal surface considerably reduce the activation energy of adsorption and dissociation, and thus enhance the chemisorption rate of oxygen molecules which instantly dissociate into neutral atoms [51]. Secondly, numerous grain boundaries and dislocations serve as efficient diffusivity paths for interstitial gaseous atoms and accelerate oxygen diffusion into the lattice of a metal [43]. Meanwhile, with additional kinetic energy from ultrasonic impact [22], the oxygen atoms readily penetrate through the oxide layer towards the oxide-metal interface. Finally, the grain boundaries and dislocations with loose atomic arrangement and high energy accelerate the transformation of ions (e.g., $O^{2-}$, $Ti^{4+}$) and electrons during oxidation, and then the preferential nucleation and crystallization of $TiO_2$ occur at grain boundaries and dislocations, i.e., numerous grain boundaries and dislocations as well as the additional ultrasonic energy promote the nucleation of oxide. Besides, the growth of oxide is inhibited due to fast heating of IHT, leading to refinement of induction heated oxide coating on titanium. In addition, IHT promotes crystallization of amorphous oxide, therefore submicro-/nano-scale morphological coatings with rutile and anatase $TiO_2$ are obtained on titanium substrate.

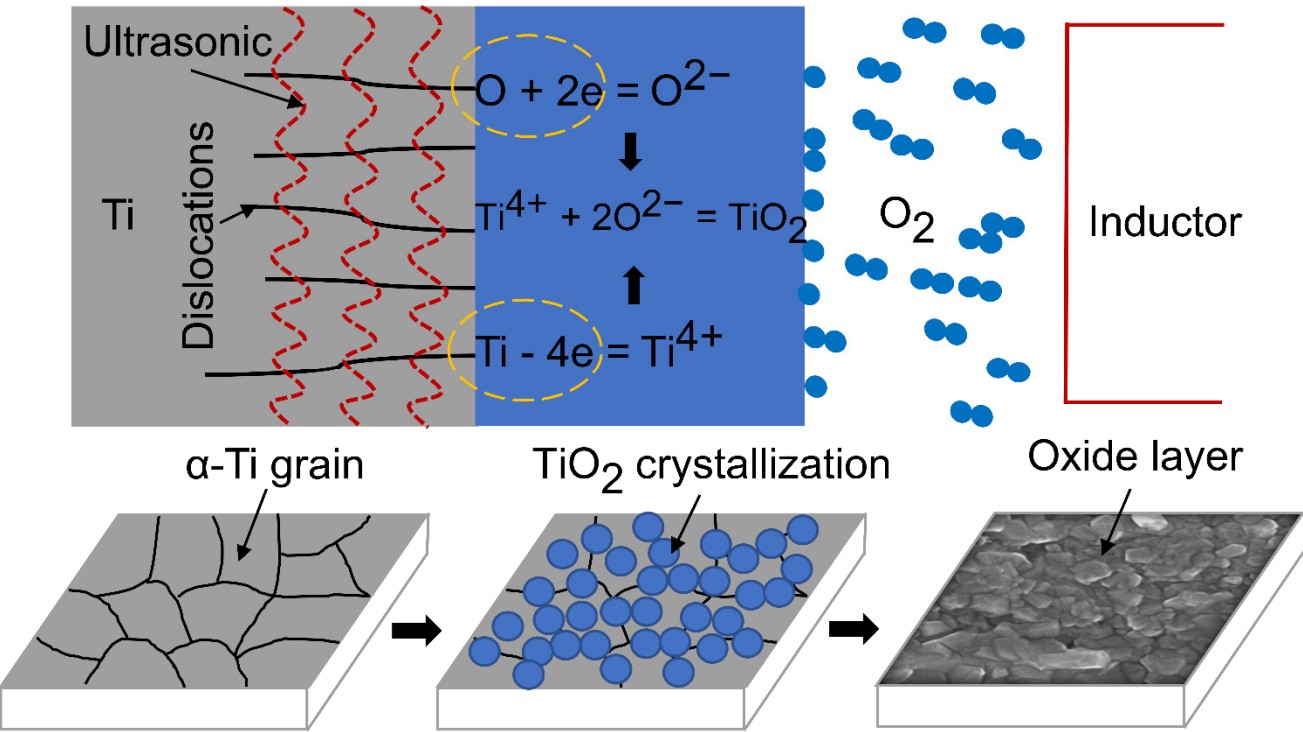

**Figure 9.** Schematic diagram of ultrasonic induced refinement of induction heated oxide coating on titanium.

## 4. Conclusions

In this work, the submicro- or nano- scale morphological oxide coatings are prepared on the cp-Ti surfaces by ultrasonic-assisted IHT for 13 s, 20 s and 25 s. The oxide coatings mainly contain the rutile/anatase TiO$_2$. Especially, the TiO$_2$ grain size significantly decreases due to the ultrasonic impact. The average sizes of TiO$_2$ particles decrease from 60 nm, 70 nm and 150 nm in the control to 10 nm, 40 nm and 100 nm in the ultrasonic-assisted samples after IHT for 13 s, 20 s and 25 s, respectively. Compared with the machined titanium, the hydrophilicity and hardness of titanium samples are improved after ultrasonic-assisted IHT. This study provides an alternative method to refine the nanoscale oxidation coatings on titanium substrates, which is promising for the further clinical development of titanium-based biomaterials. Biological behaviors of this surface, including protein adsorption, cell and tissue response, et al., need to be further investigated.

## 5. Patents

ZL 2019 1 0281286.8.

**Author Contributions:** Conceptualization, Y.L.; methodology, G.X.; validation, H.G.; investigation, X.C.; resources, M.Y.; writing—original draft preparation, H.G.; writing—review and editing, B.L.; supervision, C.C.; project administration, Y.L.; funding acquisition, B.L. and G.X. All authors have read and agreed to the published version of the manuscript.

**Funding:** This research was funded by Natural Science Foundation of Shandong Province, China (Grant No. ZR2017MEM014); Natural Science Foundation of Jiangsu Province, China (Grant No. BK20190208); Postdoctoral Research Foundation of China (Grant No. 2019M652378); and National Natural Science Foundation of China (Grant No. 52002223).

**Institutional Review Board Statement:** Not applicable.

**Informed Consent Statement:** Not applicable.

**Data Availability Statement:** Not applicable.

**Conflicts of Interest:** The authors declare no conflict of interest.

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
