# Peer review of "Ultrasonic Induced Refinement of Induction Heated Oxide Coating on Titanium"

_coatings, doi:10.3390/coatings11070812_

Round 1
Reviewer 1 Report
The article entitled "Ultrasonic induced refinement of induction heated oxide coating on titanium" is technically sound and covers the scope of Coatings.
The paper can be accepted after some minor revisions.
1- Please separate each peak in the graph or make stacks in Figure 2 for clear comparison and reader`s ease.
2- Please write significant values from results in the conclusion to support the claims in the introduction.
3- Add more references to show the attributes of ultrasonic energy such as https://doi.org/10.1016/j.ultsonch.2016.01.020
4-Authors should mention the temperature condition separately in section 2.2, and how it was maintained?
5- Describe the role of change in frequency with suitable references.
6- Authors related water contact angle to cell adhesion, thus are recommended to provide some latest references e.g.
https://doi.org/10.1016/j.msec.2020.111638 https://doi.org/10.1016/j.mtcomm.2021.102113
7- The manuscript is lacking references and some proper reasoning in the results and discussion part of the manuscript, the connection between the results is lost at some places. Authors are advised to check carefully and revise it.
Reviewer 2 Report
- On page 3, lines 110-112, the authors wrote: "The 3D topography and roughness were measured using laser scanning confocal microscopy...". Please provide more details of the surface topography measurement, including measurement methods (horizontal or vertical scanning), measurement area, numbers of measurements, etc.
- The first remark also applies to other studies.
- In addition, the authors wrote about the analysis of surface roughness parameters Sa and Sq. On what basis and why were these parameters selected for the surface characteristics? Page 7, figure 5. How many measurements have been made to determine the average value of the surface roughness parameters (Sa and Sq)? The figure shows that these were single measurements, so where did the mean values come from?
- Page 10, paragraph 3.6. The title of the paragraph is "Mechanism". What mechanism? What is this mechanism about? - the title "Mechanism of ... (of something)" should be clarified."
Reviewer 3 Report
The manuscript discussed the possibility of formation and characterization of TiO2 films on Ti substrate by ultrasonically assisted induction heating treatment (IHT). It was reported an improvement of the wettability and hardness. My specific comments are given below:
- Lines 138-140: The authors mentioned that a change in the peak intensity at the ultrasonically assisted sample is observed, which is attributed to the grain refinement. However, the grain size can be expressed by the Scherrer equation where the FWHM and the peak position of the considered diffraction maximum are used. The changes in the peak intensities could be attributed to transformations in the preferred crystallographic orientation.
- It is not clear for the readers why the increase in the IHT time results in an increase in the TiO2 particle size (line 160). The authors should discuss the results concerning the technological conditions and processes occurring during the IHT.
- The authors should clearly define the meaning of Sa and Sq parameters. The importance of surface roughness for the manufacturing of implants is not discussed. The authors should add relevant information. The authors can provide a correlation between the results obtained by laser scanning confocal microscopy and the wettability.
- Micro-hardness measurements: The authors should provide information about the penetration depth of the indenter. It is not clear whether the substrate influences the results or not. If yes, what is its influence? Usually, the improvement in the micro-hardness is related to the formation of a finer microstructure. However, the results obtained here show an opposite trend (i.e. the statement at line 160, and corresponding results from figure 3 are not in agreement with the higher hardness of the specimens treated with IHT for 20 seconds). The authors should add relevant information and discussion. Also, information about the importance of the hardness for the manufacturing of implants should be provided.
Round 2
Reviewer 3 Report
Thank you for clarifying the questionable points and providing extended information!